# Cerebrospinal Fluid Concentrations of Meropenem and Vancomycin in Ventriculitis Patients Obtained by TDM-Guided Continuous Infusion

**DOI:** 10.3390/antibiotics10111421

**Published:** 2021-11-20

**Authors:** Christoph Tiede, Ute Chiriac, Daniel Dubinski, Florian J. Raimann, Otto R. Frey, Anka C. Röhr, Anna Wieduwilt, Michael Eibach, Natalie Filmann, Christian Senft, Kai Zacharowski, Volker Seifert, Jan Mersmann

**Affiliations:** 1Department of Anesthesiology, Intensive Care Medicine and Pain Therapy, Goethe University, 60590 Frankfurt am Main, Germany; christoph-tiede@web.de (C.T.); florian.raimann@kgu.de (F.J.R.); anna-wieduwilt@web.de (A.W.); kai.zacharowski@kgu.de (K.Z.); 2Department of Pharmacy, University Hospital of Heidelberg, 69120 Heidelberg, Germany; ute.chiriac@med.uni-heidelberg.de; 3Department of Neurosurgery, Goethe University, 60528 Frankfurt am Main, Germany; daniel.dubinski@med.uni-rostock.de (D.D.); michael.eibach@kgu.de (M.E.); christian.senft@med.uni-jena.de (C.S.); volker.seifert@kgu.de (V.S.); 4Department of Pharmacy, Heidenheim General Hospital, 89522 Heidenheim, Germany; Otto.Frey@Kliniken-Heidenheim.de (O.R.F.); Anka.Roehr@Kliniken-Heidenheim.de (A.C.R.); 5Institute of Biostatistics and Mathematical Modeling, Goethe-University, 60590 Frankfurt am Main, Germany; filmann@med.uni-frankfurt.de

**Keywords:** meropenem, vancomycin, therapeutic drug monitoring, ventriculitis, pharmacokinetics, critical care, cerebrospinal fluid

## Abstract

Effective antibiotic therapy of cerebral infections such as meningitis or ventriculitis is hindered by low penetration into the cerebrospinal fluid (CSF). Because continuous infusion of meropenem and vancomycin and routine therapeutic drug monitoring (TDM) have been proposed to optimize antimicrobial exposure in ventriculitis patients, an individualized dosing strategy was implemented in our department. We present a retrospective analysis of meropenem and vancomycin concentrations in serum and CSF in the first nine ventriculitis patients treated with continuous infusion and TDM-guided dose optimization aiming at 20–30 mg/L. Median initial dosing was 8.8 g/24 h meropenem and 4.25 g/24 h vancomycin, respectively, resulting in median serum concentrations of 21.3 mg/L for meropenem and 24.5 mg/L for vancomycin and CSF concentrations of 3.4 mg/L for meropenem and 1.7 mg/L for vancomycin. Median CSF penetration was 15% for meropenem and 7% for vancomycin. With initial dosing, all but one patient achieved CSF concentrations above 1 mg/L. Dose adjustment according to TDM ensured sufficient CSF concentrations in all patients within 48 h of treatment. Given the limited penetration, continuous infusion of meropenem and vancomycin based on renal function and TDM-guided dose optimization appears a reasonable approach to attain sufficient CSF concentrations in ventriculitis patients.

## 1. Introduction

Acute subarachnoid hemorrhage, intraventricular bleeding, tumors of the brain stem, and other acute intracranial pathologies may require insertion of external ventricular drains to manage hydrocephalus and monitor intracranial pressure. However, introduction of a plastic catheter into the cerebrospinal fluid (CSF) space holds the potential for intracerebral infections such as ventriculitis and/or meningitis. Ventriculitis is the most frequent complication in these patients with significant morbidity, prolonged ICU and hospital length of stay, and higher costs [1]. 

The spectrum of pathogens in ventriculitis is characterized mainly by common skin microorganisms, which are mostly Gram-positive [2]. The Infectious Diseases Society of America therefore recommends vancomycin in combination with meropenem (or ceftazidime) for the initial treatment of infections after neurosurgery or infections related to external ventricular drains [3]. However, achieving and maintaining appropriate concentrations at the target site of infection is a significant challenge for critical care physicians. In ventriculitis, CSF-penetration of vancomycin as well as of meropenem is hindered by the blood-CSF-barrier [4]. Furthermore, pathophysiological changes in volume of distribution and augmented renal clearance alter pharmacokinetics in critically ill patients compared to what is observed in other patient groups [5,6,7].

Recent data show a highly variable penetration of vancomycin and meropenem into the CSF in ventriculitis patients [8,9,10,11]. Data from Blassmann et al. [9,10] suggest that traditional meropenem dosing (3 × 2 g administered as intermittent infusion) as well as traditional vancomycin dosing (2 × 1 g administered as intermittent infusion) do not achieve CSF concentrations above the minimum inhibitory concentration (MIC) in the majority of patients. High daily doses of up to 20 g meropenem or rather up to 8 g vancomycin and/or continuous infusion are proposed to ensure sufficient concentration. Moreover, continuous infusion of beta-lactam antibiotics is suggested to maximize bacteriological and clinical response by maintaining concentrations throughout the dosing interval [12] and continuous infusion of vancomycin decreases nephrotoxicity [13]. Since sparse data on penetration into the CSF of vancomycin and meropenem are available during continuous infusion, therapeutic drug monitoring (TDM) might be a reasonable approach to control target antimicrobial exposure and to avoid subtherapeutic concentrations [8,14,15]. Therefore, continuous infusion and a routine TDM of meropenem and vancomycin was implemented in ventriculitis patients at our department to overcome the difficulties in attaining antimicrobial target concentrations in the CSF. 

The present study retrospectively analyses the first nine patients suffering from ventriculitis, who received continuous infusion of meropenem and vancomycin based on renal function and TDM-guided dose optimization at our department. The objective was to evaluate meropenem and vancomycin concentrations in serum and CSF.

## 2. Results

Overall, 11 patients, all suffering from subarachnoid hemorrhage, have been identified as having received continuous dosing and routine TDM. Two of these had to be excluded, because microbiological analysis retrieved other sources of infection (i.e., central venous catheter infection) with consecutive changes in antibiotic treatment. Median duration of antibiotic therapy was 13 days (8; 18). All patients underwent operative revision of the external ventricular drain within the first few hours after diagnosis and at the start of antibiotic therapy, which included removal of the drain and replacement with a new external ventricular drain. An overview of patient characteristics is given in Table 1. Infection parameters C-reactive protein (CRP) and CSF leukocyte count decreased within 7 days after initiation of antibiotic therapy in all patients (Appendix A).

### 2.1. High Doses Required Due to Low Serum Creatinine Values in Neurosurgical Patients

Initial dosing yielded a median dose of 8.8 g/24 h (6.5; 13.0) meropenem and 4.25 g/24 h (3.5; 7.0) vancomycin for the first 24 h (Figure 1A). This equals 114.3 mg/kg bodyweight/24 h, and 57.2 mg/kg bodyweight/24 h, respectively (Figure 1B). The dose remained unchanged on day 2, with the exception of the initial 1 g bolus, until results from the 24 h time-point were retrieved. It is noteworthy that no renal impairment could be detected despite high-dose vancomycin therapy. Dose adjustment after 48 h was realized in three of nine patients for meropenem and in four of nine patients for vancomycin according to serum target concentration. The dose was reduced in one of nine patients for vancomycin, and the dose was increased in three of nine patients for both vancomycin and meropenem. Further adjustments after the second concentration analysis or later were necessary in four of nine patients, in three of which the dose was decreased, while one patient required higher doses after the second analysis.

### 2.2. Meropenem and Vancomycin Concentrations in CSF within Target Range after Dose Adjustment

Median C_CSF_/C_serum_ ratio was 0.15 (0.06; 0.33) for meropenem and 0.07 (0.02; 0.50) for vancomycin. Table 2 shows meropenem and vancomycin concentrations in serum and CSF of each patient 24 h after infusion start (Table 2). As compared with meropenem concentrations (3.4 [0.8; 9.8] mg/L), observed vancomycin concentrations after the first 24 h of treatment were more variable in CSF (1.7 [0.5; 12.5] mg/L). For meropenem, patients 6 and 7 showed low C_CSF_/C_serum_ ratios as well as C_serum_ < 20 mg/L, resulting in insufficient CSF concentrations. For vancomycin, only patient 6 showed an insufficient CSF concentration. After dose adjustment, however, target CSF concentration was achieved in all patients. Figure 2 and Figure 3 depict the course in serum and CSF concentrations during the first 7 days of treatment.

### 2.3. Initial Dosing Recommendation Based on Nomograms

Values estimated in the first simulation using the model developed for intermittent application did not correlate with the observed values in the present study. We therefore added a bolus of 1 g of meropenem or vancomycin (i.e., X1 was set to 1 g at time-point zero), set the infusion rate [R(t)] of the drug in relation to the volume of the central compartment, and fitted the model to the study data. These changes lead to the following equation:$$\frac{\mathrm{dX}1}{\mathrm{dt}}=\frac{R\left(t\right)}{V_{c}}-\frac{\mathrm{CL}}{V_{c}}\times X1-k_{\mathrm{cp}}\times X1-k_{\mathrm{cb}}\times X1+k_{\mathrm{pc}}\times X2+k_{\mathrm{bc}}\times X3$$
where X1 describes the quantity of the drug in the central compartment, X2 stands for the quantity of the drug in the peripheral compartment, and X3 relates on the quantity in the CSF. CL is the clearance of the drug from the central compartment, V_c_ is the volume of the central compartment, and k_bc_, k_cb_, k_pc_, and k_cp_ are distribution constants, which represent the transition between the three compartments. A schematic representation of the structural model is shown in Appendix A. For meropenem, fitted values for each constant were 0.0248 (0.0055; 0.0556; k_cb_), 0.1642 (0.0421; 0.2977; k_bc_), 14.6389 (8.4759; 28.5090; CL), and 13.7958 (5.2776; 14.7230; V_c_). For vancomycin, using the present pharmacokinetic model parameter adjustment failed to mimic vancomycin distribution between compartments.

Measured creatinine clearance is a good predictor of meropenem clearance calculated by the pharmacokinetic model with a rank correlation coefficient according to Spearman of ρ = 1 (Figure 4). The adjusted model allowed for estimation of the daily doses of continuous infusion meropenem required to yield concentrations of 2, 3, or 4 mg/L in CSF given different creatinine clearance values. These data were plotted, and regression analysis retrieved nomograms, as depicted in Figure 5.

## 3. Materials and Methods

### 3.1. Study Design

This retrospective analysis was conducted according to the guidelines of the Declaration of Helsinki and approved by the ethics committee of Goethe University, Frankfurt am Main, Germany (# 4/17).

Patients included were diagnosed with ventriculitis secondary to subarachnoid hemorrhage, received continuous infusion of meropenem and vancomycin, and received TDM measurements between January 2016 and March 2017.

Initial dosing of meropenem and vancomycin was calculated by an algorithm published by Pea et al. [16] and adapted by Minichmayr et al. [7]. For vancomycin, serum concentrations of 20 to 30 mg/L were targeted to achieve the pharmacodynamic target of the area under the concentration curve divided by the MIC to be above 400 for susceptible Staphylococcus aureus isolates [17]. For meropenem, serum concentrations of 20 to 30 mg/L were targeted corresponding 10 to 15 times the non-species related breakpoints of the European Committee on Antimicrobial Susceptibility Testing’s MIC90 data (http://www.eucast.org/clinical_breakpoints, accessed on 16 December 2015). All patients received an initial bolus of 1 g meropenem and 1 g vancomycin as 60 min infusion prior to continuous infusion. 

Blood and intraventricular CSF samples were taken 24 h and 72 h after the start of continuous infusion. Further sampling was performed as needed (Figure 6). For transport, samples were frozen immediately after preparation. Results were retrieved within 24 h. If necessary, antibiotic dosages were adjusted according to TDM results.

Infection and renal function parameters, fluid intake, and urinary excretion were extracted from the patients’ medical record. Creatinine serum concentrations were measured daily, and creatinine clearance was calculated using the Cockcroft–Gault equation [18].

### 3.2. Bioanalytical Methodology 

The analyses were performed in the laboratory of the Pharmacy Department of the Hospital of Heidenheim. Serum and CSF concentrations of meropenem were analyzed using a validated high-performance liquid chromatography (HPLC) assay with ultraviolet detection [19]. The assay was linear from 1–30 mg/L in serum and 0.5–5 mg/L in CSF with a relative standard deviation for intra- and interday precision and accuracy of <5% at high, medium, and low concentrations. The limit of quantification was 0.5 mg/L for serum samples and 0.2 mg/L for CSF samples. Serum and CSF concentrations of vancomycin were analyzed using an in vitro chemiluminescent micro particle immunoassay (ARCHITECT iVancomycin assay, Abbott; measuring range: 0.24 mg/L–100.00 mg/L). For CSF concentrations of vancomycin, spiked CSF samples were carried out on a regular basis in addition to the routine validation process. Only total concentrations (bound plus unbound) of vancomycin and meropenem were measured. 

### 3.3. Statistics

Statistical analysis was conducted using R version 3.2.4 (2016, The R Foundation for Statistical Computing). Besides standard methods for descriptive statistics (i.e., calculation of medians, quartiles, and boxplots) we used a robust local regression method, where outliers are iteratively identified and down-weighted (package locfit, locfit.robust).

Total and cumulative doses, serum, and CSF concentrations as well as distribution constants are given as median (min, max). 

The first step of the statistical analysis was to compare measured concentrations of meropenem and vancomycin with simulated values using the pharmacokinetic model and calculation developed by Blassmann et al. [9]. This was followed by adjustment of the parameters with the R packages FME, deSolve, rootSolve, and coda (2016, The R Foundation for Statistical Computing). A new simulation with the fitted parameters for meropenem was carried out, by using the package deSolve. The relationship between the clearance of meropenem and the clearance of creatinine is shown in a scatterplot. In addition, we calculated the rank correlation coefficient according to Spearman. 

Having adjusted the distribution constants for each individual patient, meropenem doses for three different CSF target concentrations were estimated. These estimates were plotted to create a nomogram for initial dosing of meropenem until TDM results are retrieved. 

## 4. Discussion

In this study, we retrospectively evaluated vancomycin and meropenem concentrations in serum and CSF of ventriculitis patients receiving continuous infusion and routine TDM. We found that all but one patient achieved CSF concentrations of meropenem and vancomycin above 1 mg/L with initial dosing based on renal function. Dose optimization according to TDM ensured sufficient CSF concentrations in all patients within 48 h. However, there was considerable variation in serum and CSF concentrations as well as resultant CSF/serum ratios. These findings are concordant with those derived from previous studies in which researchers have also described large interindividual variability [10]. Therefore, our study suggests routine TDM of meropenem and vancomycin in CSF further on to avoid underexposure potentially resulting in treatment failures. 

Penetration of the blood–CSF barrier for most antibiotic drugs occurs, if at all, slowly, resulting in much lower concentrations in the CSF as compared with serum concentrations. Drug as well as disease factors, such as the presence of meningeal inflammation and the integrity of the blood–CSF barrier, are important to consider when devising dosing strategies in patients with intracerebral infection [1]. During meningeal inflammation disruption of cell–cell contacts increase blood–CSF barrier permeability and decrease CSF outflow resulting in decreased antibiotic elimination from the CSF [20]. In ventriculitis, the meninges are typically normal or only minimally inflamed [20]. Serum-to-CSF ratios range between 1 and 18% [1,10,11] for vancomycin and between 5 and 9% for meropenem in ventriculitis patients [1,10,11], compared with reported serum-to-CSF ratios of between 6 and 81% for vancomycin and 21 and 39% for meropenem in patients with bacterial meningitis [1,11]. Hence, the poor penetration into the CSF described in previous studies led us to aim at vancomycin and meropenem serum concentrations of 20–30 mg/L. Regarding concentration–toxicity relationships of meropenem, Imani et al. recently reported that meropenem threshold concentrations of Cmin  =  64.2 mg/L and Cmin  =  44.5 mg/L in serum were associated with a 50% increased risk of a neurotoxic or nephrotoxic event during intermittent dosing, respectively [20]. The upper threshold of 30 mg/L investigated in the present work represents a concentration for which higher values will not result in increased efficacy, but for which toxicity is more likely and thus seems a reasonable limit.

Higher than standard doses of meropenem and vancomycin were needed to reach target serum concentrations of 20–30 mg/L. Recommended daily doses for meropenem are 6 g in adults [21], while within the first 24 h, a median dose of 8.8 g (114.3 [76.5; 142.9] mg/kg bodyweight) was used in our adult population to attain target serum and CSF concentrations. Recommended daily doses for vancomycin in adults are 2 g (or 30–60 mg/kg bodyweight) [21], as compared to a median dose of 4.25 g (57.2 [41.2; 68.6] mg/kg bodyweight) within the first 24 h in this study. However, renal function in our study was greater than normal creatinine clearance (>100 mL/min). This is common in neurocritical care patients [5] and may lead to higher doses to achieve a target concentration in serum similar to that observed in other critically ill patients. Augmented renal clearance has previously been shown as an independent predictor of not achieving sufficient concentrations with standard doses [22]. Nomograms based on ventriculitis patient data might help estimate antibiotic doses required to achieve CSF concentrations at the beginning of therapy, although nomograms should not preclude the use of TDM. A nomogram of vancomycin could, however, not be generated in the present study. Higher inter-individual variation compared with meropenem, together with significant intra-individual variation during the course of the infection (not shown), hindered successful modelling of vancomycin pharmacokinetics. Moreover, parameters k_cp_ and k_pc_ were not adjusted, because there were no measured values for the peripheral compartment. Median penetration into the CSF in this study was higher than that reported in ventriculitis patients with intermittent infusion (meropenem: 15% vs. 9%; vancomycin: 7% vs. 3%) [9], and is comparable to the penetration of meropenem and vancomycin administered as continuous infusion reported by Mader et al. (meropenem: 15 vs. 18%, vancomycin: 7 vs. 20%) [8]. For vancomycin, a higher variability compared to meropenem was observed in the present study, also reported by Mader et al. [8] However, all studies were relatively small, which may have hampered robust estimates of the extent of pharmacokinetic variability. Furthermore, pharmacokinetic data were not adjusted for protein binding, because we measured the total concentration in serum and CSF. Protein binding is not relevant for meropenem, and CSF concentrations might be free concentrations due to negligible protein levels in CSF compared to serum. Altered protein binding of vancomycin in serum might influence the free concentration penetrating into CSF. Further studies are needed, including a higher number of patients as well as systematic CSF and serum concentration sampling.

The present study was not designed for efficacy evaluation. More work is required to better understand pharmacodynamic targets at the site of infections for patients with ventriculitis. It is unclear which pharmacokinetic–pharmacodynamic target should be used to optimize the outcome for intracerebral infections [1]. Thus, we cannot draw conclusions about the role of TDM as a potential to improve efficacy. However, considering that MIC breakpoint for susceptibility were exceeded and treatment was successful in all patients, it is reasonable to assume that the application of TDM may help in early identification of patients underexposed to antimicrobial therapy, having minimum drug concentrations below the MIC. In these patients, optimization of drug dose guided by TDM would eventually contribute to improve the response to antimicrobial therapy.

## 5. Conclusions

These data show that continuous infusion of vancomycin and meropenem based on renal function and TDM-guided dose optimization ensured sufficient CSF concentrations within 48 h of treatment. The results suggest that this approach appears feasible and effective in cases of ventriculitis, given the limited penetration of meropenem and vancomycin into CSF. However, data are sparse and higher than standard doses are necessary. This approach should be evaluated with a larger number of patients in a prospective study in the future.

## Figures and Tables

**Figure 1 antibiotics-10-01421-f001:**
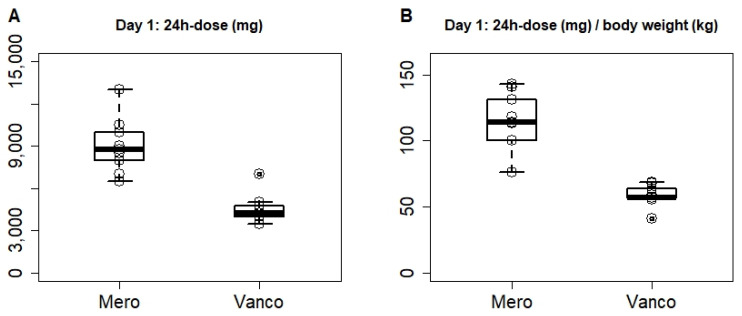
Applied dose of meropenem (Mero) and vancomycin (Vanco) over the first 24 h (total **A**, or in relation to body weight **B**). Depicted are the median, and 50% quartile ranges; dots depict individual patient values (n = 9).

**Figure 2 antibiotics-10-01421-f002:**
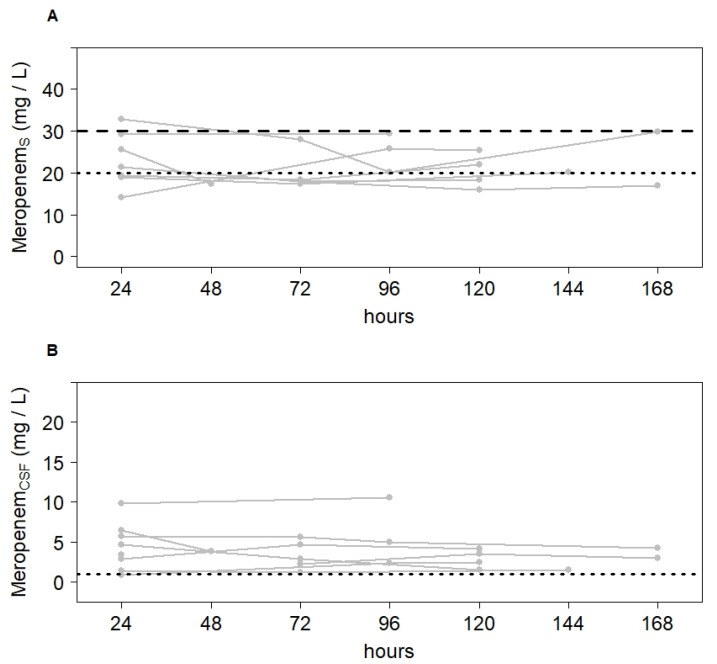
Variation of meropenem concentrations in serum (**A**) and CSF (**B**) over the first 7 days of treatment. Narrow lines depict the course of every individual patient (n = 9). The dotted line depicts the lower and the dashed line the upper limit of the target concentration.

**Figure 3 antibiotics-10-01421-f003:**
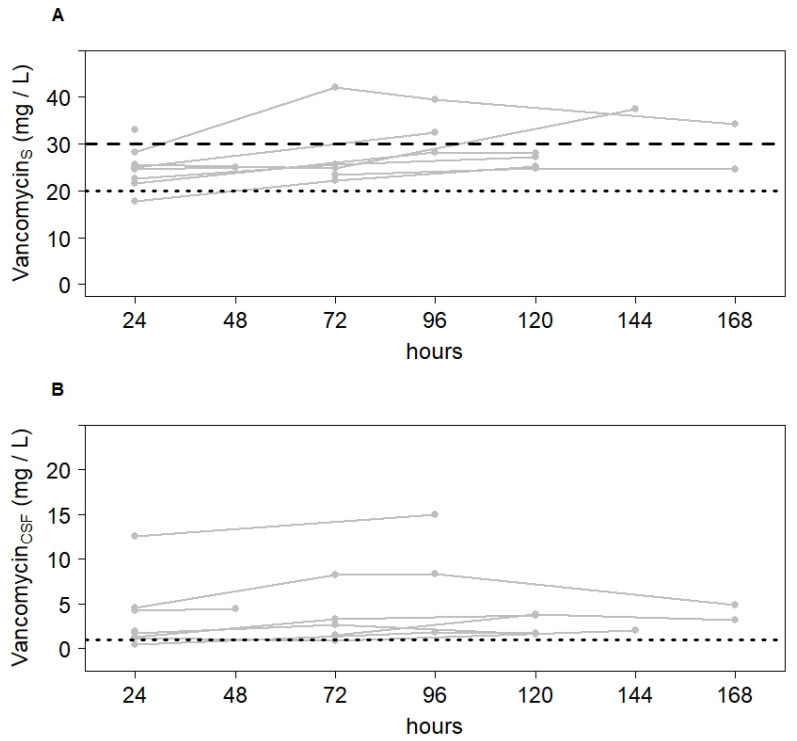
Variation of vancomycin concentrations in serum (**A**) and CSF (**B**) over the first 7 days of treatment. Narrow lines depict the course of every individual patient (n = 9). The dotted line depicts the lower and the dashed line the upper limit of the target concentration.

**Figure 4 antibiotics-10-01421-f004:**
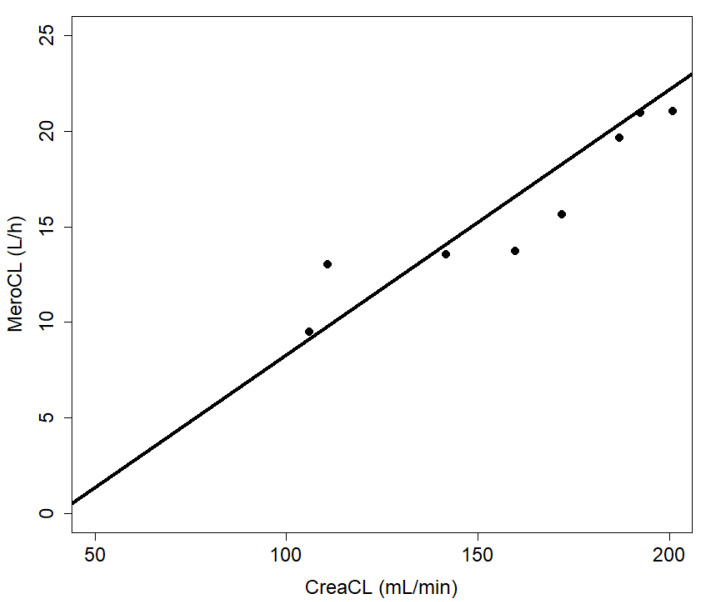
Meropenem and creatinine clearance. The clearance of meropenem (MeroCL) and creatinine (CreaCL) show a positive correlation with a rank correlation coefficient according to Spearman of ρ = 1.

**Figure 5 antibiotics-10-01421-f005:**
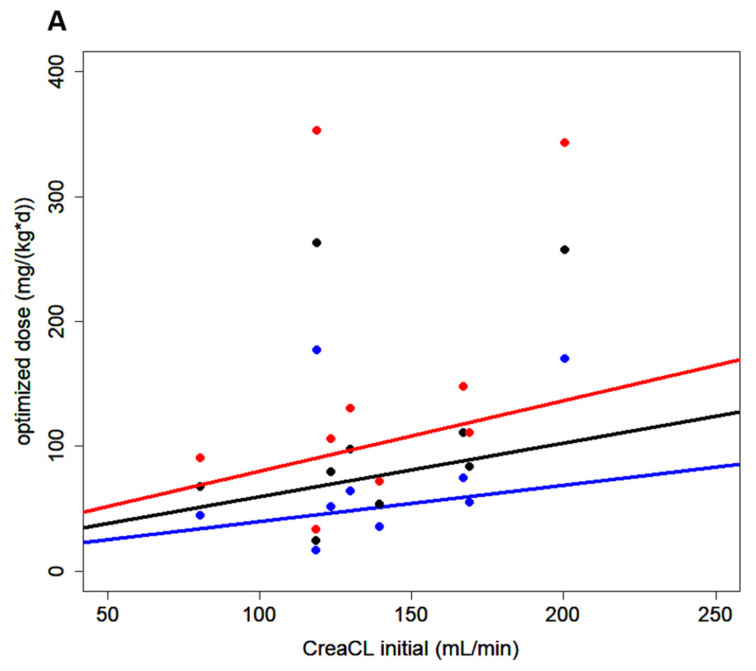
Nomogram for initial dosing of meropenem. Nomogram depicting the daily doses of continuous infusion meropenem required to achieve specific target concentrations in CSF (2, 3, and 4 mg/L) given different creatinine clearance values (CreaCL, calculated according to Cockcroft and Gault). Blue, black, and red lines and dots represent calculated doses for target CSF concentrations of 2, 3, or 4 mg/L, respectively.

**Figure 6 antibiotics-10-01421-f006:**
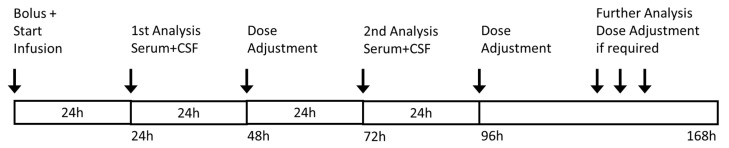
Treatment regimen. Serum and CSF concentrations were analyzed after 24 and 72 h with consecutive dose adjustment after 48 or 96 h from the start of infusion. Further analyses and dose adjustment were only performed where necessary.

**Table 1 antibiotics-10-01421-t001:** Patient characteristics.

Patient Characteristics	
Age, years, median (min, max)	53 (29, 84)
Weight, kg, median (min, max)	75 (62, 110)
Height, cm, median (min, max)	180 (160, 190)
Body-Mass-Index, kg/m^2^, median (min, max)	24.5 (22.9, 32.1)
Sex, male/female	7/2

**Table 2 antibiotics-10-01421-t002:** Serum and CSF concentrations in each patient 24 h after infusion start.

		Meropenem	Vancomycin
**Patient 1**	C_serum_ [mg/L]	25.7	24.5
	C_CSF_ [mg/L]	6.4	4.2
	C_CSF_/C_serum_	0.25	0.17
**Patient 2**	C_serum_ [mg/L]	29.3	25.0
	C_CSF_ [mg/L]	9.8	12.5
	C_CSF_/C_serum_	0.33	0.50
**Patient 3**	C_serum_ [mg/L]	21.3	22.5
	C_CSF_ [mg/L]	4.6	1.7
	C_CSF_/C_serum_	0.22	0.07
**Patient 4**	C_serum_ [mg/L]	25.6	33.0
	C_CSF_ [mg/L]	3.4	1.9
	C_CSF_/C_serum_	0.13	0.06
**Patient 5**	C_serum_ [mg/L]	19.3	17.8
	C_CSF_ [mg/L]	2.9	1.2
	C_CSF_/C_serum_	0.15	0.07
**Patient 6**	C_serum_ [mg/L]	14.1	21.5
	C_CSF_ [mg/L]	*0.8*	*0.5*
	C_CSF_/C_serum_	0.06	0.02
**Patient 7**	C_serum_ [mg/L]	18.9	25.7
	C_CSF_ [mg/L]	1.4	1.2
	C_CSF_/C_serum_	0.07	0.05
**Patient 8**	C_serum_ [mg/L]	17.9	23.3
	C_CSF_ [mg/L]	2.2	1.5
	C_CSF_/C_serum_	0.12	0.06
**Patient 9**	C_serum_ [mg/L]	32.9	28.2
	C_CSF_ [mg/L]	5.7	4.5
	C_CSF_/C_serum_	0.17	0.16
**Median**	C_serum_ [mg/L]	21.3	24.5
	C_CSF_ [mg/L]	3.4	1.7
	C_CSF_/C_serum_	0.15	0.07

## Data Availability

The data presented in this study are available on request from the corresponding author. The data are not publicly available due to institutional data protection policy.

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
