# Peer review of "Cerebrospinal Fluid Concentrations of Meropenem and Vancomycin in Ventriculitis Patients Obtained by TDM-Guided Continuous Infusion"

_antibiotics, 2021, doi:10.3390/antibiotics10111421_

Round 1

Reviewer 1 Report

The authors, by a retrospective analysis of serum/CSF vancomycin and meropenem concentration, aimed to preliminary demonstrate the feasibility and effectiveness of continuous infusion of these drugs based on renal function and TDM analysis.  The manuscript is clear and the results are reliable. The data can support the hypothesis; even considering some limitations correctly identified by the authors, i.e. the need to extend the number patients and to obtain data regarding concentration in the peripheral compartment for refining the PK model.  

However, very few points should be considered:

1. Which is the rationale for the 1g bolus infused in the first 60 minutes at the beginning of the treatment? Is this initial bolus needed to achieve serum and, mainly, CSF effective concentration? Could a higher “loading dose” lead to toxic concentration?

2. Always regarding the initial 1g bolus for both drugs, could the authors better explained in which ways it has been considered for the equation reported at page 5?

3. Page 8 line 227. There is a typing error: ‘Infefection’ instead ‘infection’.

Author Response

The authors, by a retrospective analysis of serum/CSF vancomycin and meropenem concentration, aimed to preliminary demonstrate the feasibility and effectiveness of continuous infusion of these drugs based on renal function and TDM analysis.  The manuscript is clear and the results are reliable. The data can support the hypothesis; even considering some limitations correctly identified by the authors, i.e. the need to extend the number patients and to obtain data regarding concentration in the peripheral compartment for refining the PK model.  

However, very few points should be considered:

  1. Which is the rationale for the 1g bolus infused in the first 60 minutes at the beginning of the treatment? Is this initial bolus needed to achieve serum and, mainly, CSF effective concentration? Could a higher “loading dose” lead to toxic concentration?
  2. Always regarding the initial 1g bolus for both drugs, could the authors better explained in which ways it has been considered for the equation reported at page 5?
  3. Page 8 line 227. There is a typing error: ‘Infefection’ instead ‘infection’.

We would like to thank the reviewer for her/his thorough review of our manuscript. We have now revised our manuscript and addressed the reviewers’ concerns:

  1. We decided to use a 1g bolus before the start of continuous infusion with the assumption that high initial serum levels would more rapidly establish effective concentrations in the target compartment. This concept was derived from sepsis/ARDS concepts (“hit hard and early”), in which continuous or prolonged discontinuous dosing is regularly combined with an initial bolus to avoid a delay until concentrations above MIC are attained. In the study by Pea et al. (Ref #16), who developed a dosing nomogram for continuous meropenem dosing in critically ill patients, that was adapted for the present study, an initial bolus also was used. Whether “toxic” doses might have been reached during initial bolus has not been controlled for in our study. Since 1g of meropenem or 1g of vancomycin are regular “single shot” doses we, however, regard this to be rather unlikely.
  2. The initial bolus was added to the dose applied during the first 24 hrs of infusion until the first sampling time-point. To address this X1 in the equation was set to 1g at time-point zero. We have added this information. Please refer to our revised manuscript.
  3. The typing error has been corrected.

Reviewer 2 Report

This study/paper is very significant and represents critical antibiotic therapy optimization by individual dosing strategy for the cerebral infections.

Paper is adequately referenced and the quality/standard of reaserch work/trials conducted is high.

This paper has a definite potential to provide an optimized way of utilizing antibiotic therapy in cerebral infections and will certainly help patients with these infections. 

Author Response

Reviewer 2

This study/paper is very significant and represents critical antibiotic therapy optimization by individual dosing strategy for the cerebral infections.

Paper is adequately referenced and the quality/standard of reaserch work/trials conducted is high.

This paper has a definite potential to provide an optimized way of utilizing antibiotic therapy in cerebral infections and will certainly help patients with these infections.

We would like to thank the reviewer for her/his thorough review of our manuscript. We are grateful for the positive reception of our manuscript.

Reviewer 3 Report

Personalized medicine can only be accomplished when the drug treatment is associated with monitoring the drug concentrations in the central compartment as well as the target site of action. The authors conducted an interesting work by monitoring the concentrations of meropenem and vancomycin in both serum and CSF. However, the manuscript needed few revisions before it can be deemed publishable.

1) It is not clear why the authors used median to represent the variability across subjects. Median is a statistical measure used only for representing Tmax. For serum concentrations, I would suggest considering the arithmetic mean. Please revise the manuscript to reflect this statistical measure of central tendency.

2) The conclusion section is nothing but a mere limitation section. The manuscript does not have a true conclusion section. Please conclude the results appropriately.

3) The dose regimen for meropenem and vancomycin in the abstract does not match with the main text. Please check and correct it.

4) The introduction and discussion sections can be improved. 

Author Response

Reviewer 3

Personalized medicine can only be accomplished when the drug treatment is associated with monitoring the drug concentrations in the central compartment as well as the target site of action. The authors conducted an interesting work by monitoring the concentrations of meropenem and vancomycin in both serum and CSF. However, the manuscript needed few revisions before it can be deemed publishable.

1) It is not clear why the authors used median to represent the variability across subjects. Median is a statistical measure used only for representing Tmax. For serum concentrations, I would suggest considering the arithmetic mean. Please revise the manuscript to reflect this statistical measure of central tendency.

2) The conclusion section is nothing but a mere limitation section. The manuscript does not have a true conclusion section. Please conclude the results appropriately.

3) The dose regimen for meropenem and vancomycin in the abstract does not match with the main text. Please check and correct it.

4) The introduction and discussion sections can be improved. 

We would like to thank the reviewer for her/his thorough review of our manuscript. We have now revised our manuscript and addressed the reviewers’ concerns:

  1. We appreciate your advice. The study was relatively small to obtain robust estimates of the extent of PK variability in the described population. According to the limited number of subjects (9 patients) in this analysis, the median was chosen as the adequate measure of central tendency because it is more robust to outliers. This is a common approach to describe serum concentrations, although this is arithmetic data. You can also find this approach in many other studies investigating antibiotic concentrations e.g. Roberts JA et al. Clin Infect Dis. 2014; 58(8):1072-83; Scharf C et al. Antibiotics 2020 21;9(3):131xxx. We have also taken out the “mean line” from Figures 2 and 3, to avoid confusion.
  2. Thank you for your advice. We have revised the discussion and conclusion section.
  3. We have calculated two different daily doses: 1. Dose applied in the first 24hrs including the 1g initial bolus. 2. The daily dose when averaging the first 7 days of treatment. To improve clarity we decided, not to present the latter in the manuscript. However, in the methods section we gave this dose, while in the abstract and discussion the dose applied in the first 24 hours was given. We corrected this mistake. Please refer to our revised manuscript.
  4. We also have revised the introduction section.